



# 1   ITS_LIVE global glacier velocity data in near real time

Alex S. Gardner[1], Chad A. Greene[1], Joseph H. Kennedy[2], Mark A. Fahnestock[2], Maria Liukis[1], Luis A.
López[3], Yang Lei[4], Ted A. Scambos[5], Amaury Dehecq[6]
[1]Jet Propulsion Laboratory, California Institute of Technology, Pasadena, USA
[2]University of Alaska Fairbanks, Fairbanks, AK, USA
[3]National Snow and Ice Data Center, Boulder, CO, USA
[4]National Space Science Center, Chinese Academy of Sciences, Beijing, China
[5]University of Colorado Boulder, Boulder, CO, USA
[6]University of Grenoble Alpes, IRD, CNRS, INRAE, Grenoble INP, IGE, 38000 Grenoble, France
*Correspondence to*: Alex S. Gardner (alex.s.gardner@jpl.nasa.gov)
**Abstract.** Glaciers and ice sheets cover some 15 million square kilometres of the Earth's surface, shaping continental
landscapes and modifying climate on a global scale. Recent decades of atmospheric and oceanic warming have induced
rapid glacier loss worldwide that has caused sea level rise, flooding, changes to Earth's overall energy balance and changes
in water resources. Accounting for the total impact of glacier change requires observations on a global scale, and planning
for future change will require improved understanding of the physical controls that govern glacier change. One key factor
that dictates glacier and ice sheet loss is changes in rates of ice flow, the physics of which remain poorly constrained. Our
physical understanding of ice flow can be advanced with high resolution monitoring of glacier flow, in near real time.
Automated tracking of glacier flow from space became possible with the launch of Landsat 4 in 1982. Since then, an
increasing number of optical and radar satellite sensors have now provided a full decade of year-round, global data coverage.
This recent plethora of data has introduced new challenges for efficiently processing such large and myriad data streams, in
a standardized manner, with low latency. Here we present the NASA MEaSUREs Inter-mission Time Series of Land Ice
Velocity and Elevation (ITS_LIVE) global glacier velocity dataset, which is freely available to the public and is currently
on major release version 2.0. ITS_LIVE has computed surface velocities using every, excluding those with high cloud cover,
available image from Landsat 4 through 9 and Sentinel 1 & 2, creating a global glacier velocity record of over 36 million
image pairs dating back to 1982. The ITS_LIVE processing chain automatically performs feature tracking on more than
20,000 image pairs per day, within minutes of image availability, and will soon include data from Sentinel 1C and NISAR
satellites. This paper describes the ITS_LIVE processing chain and provides guidance for working with the cloud-optimized
velocity data it produces. All ITS_LIVE velocity data can now be accessed freely, without login credentials or any other
barriers, through https://its-live.jpl.nasa.gov/.



## 1 Introduction


In recent decades, glacier velocity observations have revealed a complex and evolving landscape of ice flow that
spans the globe and shapes the Earth's surface and human behaviour. In high latitudes, glacier flow has carved great fjords,
such as those of Greenland, where fishing and tourism along the coast represent major pillars of the country's modern
economy (Bendixen et al., 2019). Meanwhile in lower-latitude regions, glaciers can threaten local communities with flood
risk (Bazai et al., 2021; Cook et al., 2016; Rounce et al., 2017) while also providing a critical source of freshwater, increasing
their melt rates during warm periods of drought, when other sources of water run dry (Pritchard, 2019; Ultee et al., 2022).
Satellite observations show that glacier acceleration and corresponding increases in ice discharge to the ocean have raised
sea levels (IPCC, 2023; Otosaka et al., 2023), impacted ocean circulation and primary productivity (Li et al., 2024; Perner
et al., 2019), and shifted Earth's energy balance (Hansen et al., 2011; Sicart et al., 2008). Understanding and accounting for
the myriad impacts of global changes in ice dynamics have been made possible by global satellite data coverage, and as we
face a future of certain climate change, preparing for the impacts of glacier variability will require near-real time monitoring
of glacier dynamics on a global scale.
The use of satellite images to measure glacier velocity began in the 1980s with manual identification of persistent
features (e.g. crevasses) that were displaced between pairs of satellite images (e.g., Lucchitta and Ferguson, 1986; Whillans
and Bindschadler, 1988). By the 1990s, template matching algorithms (i.e. normalized cross correlation) were developed to
systematically measure displacement fields from image pairs for investigations of glacier flow (Bindschadler and Scambos,
1991; Scambos et al., 1992), and that work has since led to the development of several open-source software packages for
feature tracking, including COSI-Corr (Leprince et al., 2007), MATLAB-based ImGRAFT (Messerli and Grinsted, 2015),
Python-based PyCorr (Fahnestock et al., 2016), and the autoRIFT package (Gardner et al., 2018; Lei et al., 2021) used to
generate the ITS_LIVE data described in this paper. After years of algorithm development and remote sensing data
collection, a few major efforts have generated large-scale ice velocity mosaics that have each enabled a new wave of
advancements in glaciological observation and modelling.
One of the first large-scale ice velocity mapping projects used RADARSAT synthetic aperture radar (SAR) data
to map the flow of the Greenland Ice Sheet (Joughin et al., 2010), and soon after, multiple years of SAR data were stitched
together to create a nearly complete map of the flow of the Antarctic Ice Sheet (Rignot et al., 2011). In the years that
followed, ice-sheet-wide mosaics came available at annual (Gardner et al., 2022; Joughin, 2023a; Mouginot et al., 2017)
and subannual (Joughin, 2022, 2023b; Solgaard and Kusk, 2022) intervals, and image-pair level velocity data were made
available for the full Landsat 8 record via GoLIVE (Scambos et al., 2016) and later ITS_LIVE (Gardner et al., 2022).
Beyond the ice sheets, glacier velocity data have been generated globally for a single snapshot in time (Millan et al., 2021),
as annual mosaics (Gardner et al., 2022), and as displacement fields measured in individual image pairs from various satellite
sensors (Gardner et al., 2022; Scambos et al., 2016). The large and rapidly growing volume of remote sensing data now far





exceeds the storage and processing capabilities of any laptop computer or local workstation, meaning modern, cloud-
optimized approaches for velocity data generation and storage will be essential to usher in the next generation of
glaciological advancement (López et al., 2023). The cloud-native processing chain developed for ITS_LIVE version 2.0 is
described below.

**2 Processing System**


Due to the sheer volume of data and intensive processing needs, ITS_LIVE decided to adopt a cloud-first approach
to data processing and access. The ITS_LIVE processing chain is an AWS cloud-native application. It is composed of three
major components: (1) ITS_LIVE Monitoring that watches for new satellite image acquisitions, (2) HyP3 ITS_LIVE that
orchestrates the processing of image pairs, and (3) HyP3-autoRIFT that processes image pairs and publishes them to the
ITS_LIVE AWS OpenData S3 bucket.

**2.1 ITS_LIVE Monitoring**


The ITS_LIVE Monitoring stack (Kennedy et al., 2025a) uses an event-driven architecture to listen for new input
satellite data products to be published, find new images to use for velocity estimation, and submit qualifying image-pairs
(criteria described in Sec. 3) for processing to HyP3 ITS_LIVE (Sec. 2.2-3). Figure 1 illustrates the data flow for Landsat
Images.
For Landsat 8/9 data, ITS_LIVE monitoring subscribes to the USGS's SNS Topic which broadcasts messages
describing newly available Landsat data. All level 1, teir 1 and 2  products from Collection 2 are placed into an AWS SQS
queue, which then triggers an AWS Lambda that evaluates if the corresponding scene qualifies for processing based on the
scene's metadata in the USGS STAC catalog. Each qualifying scene is used as a reference scene, the USGS STAC catalog
is searched for qualifying secondary scenes, and each reference and secondary scene pair are submitted to HyP3 ITS_LIVE
for processing. New velocity granules are published to the its-live-data AWS S3 bucket by HyP3 ITS_LIVE. Each new
image may pair with up to 35 previous images and create 35 new velocity granules, which are typically published within 15
minutes after each new Landsat data product is broadcast.



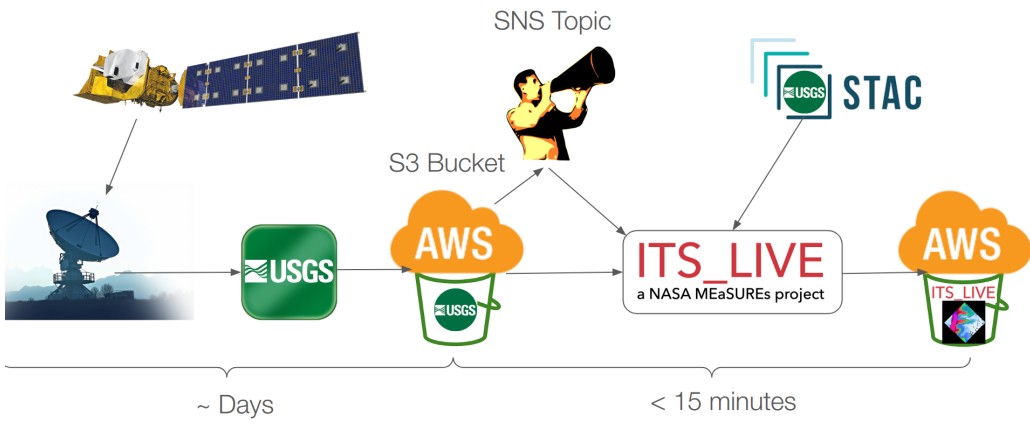


**Figure 1: Landsat data flow, from acquisition to ITS_LIVE velocity. Landsat acquisitions are downlinked, processed by the USGS ground segment, published to the USGS-Landsat AWS S3 bucket, and broadcast via an AWS SNS topic. ITS_LIVE monitoring subscribes to this topic, selects older secondary scenes to pair with the new scene, and submits the pairs for processing. Scenes are processed by HyP3 ITS_LIVE (Sec. 2.2) and new velocity granules are published to ITS_LIVE's S3 bucket. New velocity granules are typically available within 15 minutes of the corresponding new scene being broadcast. Landsat 9 Satellite rendering courtesy Northrop Gruman, USGS logos courtesy U.S. Geological Survey, and STAC logo adapted from stacspec.org.**

The Sentinel-2 data flow mimics the Landsat data flow, with the primary difference being that Sentinel-2 scenes are acquired, downlinked, and processed by ESA. Within a few hours of Sentinel-2 image availability, scenes are published by Synergize to an S3 bucket, catalogued in Element84's Earth Search STAC catalogue, and broadcast by an SNS topic in the eu-central-1 region. Once a new scene has been broadcast, we check that it qualifies for processing by using metadata from the Earth Search STAC catalogue as well as SentinelHub's RODA API since the percent data coverage is missing from the STAC catalogue. If the new scene qualifies, it is used as the reference scene, and the Earth Search STAC catalogue and RODA are used to find corresponding secondary scenes that qualify for processing. To comply with institutional policies and minimize the cost of pulling scenes across regions, we download Sentinel-2 scenes from Google Cloud. If scenes are not available on Google Cloud, we wait 8 hours and try again, repeating the check up to 3 times. Images submitted for processing may take up to 24 hours to be published in the ITS_LIVE S3 bucket from when the new scene SNS message was broadcast, meaning velocity estimates from Sentinel-2 are available within about 30 hours after publication by ESA.

The near-real-time processing chain for Sentinel-1 and NISAR resembles the Landsat processing chain, with a distinction that Sentinel-1 scenes are acquired, downlinked, and processed by ESA and NISAR scenes will be acquired, downlinked, and processed by NASA JPL. Within 24 hours of *publication* of Sentinel-1 scenes or *acquisition* of NISAR scenes, the new scenes are ingested and archived by the Alaska Satellite Facility (ASF) NASA DAAC in a private S3 bucket





and catalogued in NASA's Common Metadata Repository (CMR). CMR allows users to set up Near-Real-Time Notification
subscriptions for collections which will broadcast messages into a user-provided AWS SQS queue when new scenes are
ingested. Once a new scene has been broadcast, we will check that it qualifies for processing, query CMR for corresponding
secondary scenes, and submit pairs for processing. Processing velocity estimates for Sentinel-1 is significantly more
complex than the optical products, and this will likely hold true for NISAR products. Therefore, we expect new velocity
estimates to typically be available within 2 hours of the new scene being catalogued by NASA, or <30 hours after publication
by ESA or acquisition by NASA. It should be noted that at the time of writing, ITS_LIVE processing of Sentinel-1 was
temporarily put on hold due to the cost of processing. A major code refactoring of the SAR processing pipeline is being
undertaken to significantly reduce processing costs and to prepare ingestion of NISAR data. At the time of writing, Sentinel-
1 data has only been processed for the period 2014-2022. We anticipate low-latency SAR processing to resume by mid-
120    2025.

## 2.2 HyP3 ITS_LIVE


We utilize a custom deployment of the open-source, cloud-native ASF HyP3 processing pipeline (Hogenson et al.,
2020, Johnston et al., 2025) deployed to the us-west-2 region, which is the same region that houses USGS Landsat, NASA's
mirror of Sentinel-1, and NASA's NISAR mission products, as well as the ITS_LIVE data products. In the HyP3
architecture, storage and egress costs are minimized by bringing the compute to the data. HyP3 is built using a serverless
architecture and can easily scale to handle large processing campaigns; for ITS_LIVE v2 data, we have scaled up to 10,000
vCPUs and were able to process 25 million image pairs in a single month.
HyP3 is a user-driven, high-throughput processing pipeline. Users, or in our case, the ITS_LIVE Monitoring stack,
can request new data products through the API, which follows the OpenAPI specification and is self-documented with a
SwaggerUI, or through a Python SDK. Processing requests are tracked in an AWS DynamoDB and executed through AWS
StepFunctions, which: track the job status, runs HyP3 plugins (containers; see Sec. 2.3) via AWS Batch, and updates
processing records with information. When jobs are completed, job status, output file locations, and other relevant job
metadata is available to users via the API or SDK.

## 2.3 HyP3 autoRIFT


HyP3 autoRIFT (Kennedy et al., 2025b) is a docker container that follows the HyP3 plugin specification and is
used to process input image pairs and publish ITS_LIVE velocity granules. HyP3 autoRIFT is responsible for the end-to-
end processing workflow and contains the autoRIFT processing code (described in Sec. 3) for both the optical and radar
data, as well as a Python library that handles finding and staging necessary input data (images, DEMs, parameter files, etc.),
determining the correct processing parameters to use for a scene pair, executing the processing workflow, packaging the





autoRIFT outputs into an ITS_LIVE NetCDF data product, generating product browse and thumbnail images, and finally
uploading the products to an AWS S3 bucket.

## 3 Velocity calculations

ITS_LIVE employs a two-tiered approach (Level 2 and Level 3 products) to processing optical or SAR data streams
on common UTM or polar stereographic grids. Level 2 image pairs are processed for every available combination of satellite
images separated by fewer than a threshold number of days for each sensor, then compiled into Level 3 regional velocity
mosaics. To maximize computational efficiency and minimize distortion or loss of information that could result from
interpolation and grid transformations, ITS_LIVE developed the Geogrid algorithm
[https://github.com/leiyangleon/Geogrid] that provides a direct mapping between image coordinates (radar or optical) and
map coordinates. The algorithm allows autoRIFT to perform feature tracking in the native image coordinates that are then
directly mapped to geographic coordinates. ITS_LIVE then generates velocities on a uniform grid without the need for
resampling or interpolating, regardless of whether the data are in line-of-sight coordinates (radar) or in a native projection
that differs from the target projection. For all sensors, ITS_LIVE version 2.0 velocities are determined at the same
geographic locations on a common 120 m resolution grid.

## 3.1 Level 2 image pairs

Using a template-matching approach, displacement fields are calculated from image pairs acquired up to 546 days
apart for optical images and 12 days for radar images. Our goal is to increase the temporal span for radar images if we're
able to achieve increased processing efficiency, i.e. reduced cost. All image pairs are processed by the autonomous Repeat
Image Feature Tracking (autoRIFT) algorithm version 1.4.0, which was originally developed for Landsat imagery (Gardner
et al., 2018), has since been expanded to handle Sentinel 2 and Sentinel 1 data (Lei et al., 2021, 2022), and is now a registered
and maintained conda-forge Python package that gained wide use within the research community (e.g., Hong et al., 2022;
Kochtitzky et al., 2022; Liu et al., 2024).

### 3.1.1 Optical data from Landsat 4-9 and Sentinel 2A/B/C

ITS_LIVE continuously processes optical images from Landsat 4 (1982-1993), Landsat 5 (1984-2013), Landsat 7
(1999-present), Landsat 8 (2013-present), Landsat 9 (2021-present), Sentinel 2A (2015-present), Sentinel 2B (2017-
present), and Sentinel 2C (2024-present). Surface displacements are calculated for "same-path-row" image pairs that are
acquired from the same satellite position and look geometry and are separated in time by fewer than 546 days. To increase
data density prior to the launch of Landsat 8, images acquired from differing satellite positions (i.e. cross-path-row),
generally from crossing ascending and descending orbits, are also processed if they have a time separation between 10 and



96 days. Feature tracking of cross-path-row image pairs produces velocity fields with lower signal-to-noise due to residual
parallax from imperfect terrain correction that is largely self-cancelling in imagery acquired with the same viewing geometry
(i.e. same-path-row).
All optical images are preprocessed using a 5x5 Wallis operator to normalize for local variability in image radiance
caused by shadows, topography, and sun angle, all of which can generate spurious artifacts when applying feature tracking
to derive surface flow from optical imagery. For Landsat 4 and 5 Band 2 images, along-track artifacts introduced by the
Thematic Mapper whisk broom sensor are removed using Fourier filtering. Missing data in Landsat 7 images, introduced
after the Scan Line Corrector failure (SLC-off) in May of 2003, are filled with random noise so that they do not contribute
to the amplitude of the correlation peak used in the feature tracking.
Using autoRIFT, preprocessed same-path-row and cross-path-row pairs of images are searched for matching
features by finding local normalized cross-correlation (NCC) maxima at sub-pixel resolution by oversampling the
correlation surface by a factor of 16 using a Gaussian kernel. As a first step, a sparse grid pixel-integer NCC search (1/16
of the density of the full search grid) is used to determine areas of coherent correlation between image pairs. Results from
the sparse search guide a dense search with search centres spaced such that there is 50% overlap between adjacent template
windows. Areas of unsuccessful retrievals, as determined using a Normalized Displacement Coherence NDC filter (Gardner
et al., 2018), are searched with progressively increasing template chip sizes. Minimum and maximum acceptable template
chip sizes for each search centre are defined geographically and depend on land surface type (ice or rock), spatial gradient
of a reference velocity mapping, distance from ocean, and distance from ice edge. The data are then filtered one last time
using the NDC filter, and small data gaps are filled by interpolation. Interpolated values are indicated in each image-pair
data product as **interp_mask = 1**.
To reduce computational demand, autoRIFT employs a downstream search that centres the NCC search template
window in the search image at the expected downstream location of displacement, as determined from the reference velocity.
The NCC search radius is unique in both x- and y-directions and varies spatially. The NCC search radius is defined according
to the surface type (ice or rock), magnitude of the component reference velocity (vx, vy), and the distance from the ocean.
Ocean area is identified according to the Global Self-consistent, Hierarchical, High- resolution Geography Database
(GSHHG). In Greenland, land ice area is identified according to a data set provided by F. Paul (Bolch et al., 2013); in
Antarctica, land ice is identified according to Depoorter et al., 2013, and everywhere else land ice is determined using the
Randolph Glacier Inventory Release 6.0. Rock is defined as neither ocean nor land ice.

### 3.1.2 SAR data from Sentinel 1A-1B

ITS_LIVE continuously processes "same-path-frame" SAR images from Sentinel 1A (2014-present) and Sentinel
1B (2016-2021), separated by 12 days or fewer. When applied to SAR imagery, autoRIFT generates a rotation matrix that
allows derivative surface velocities to be generated from two Level 2 ITS_LIVE granules, one ascending and one



descending, using only range offsets that are significantly more precise than azimuth offsets (Joughin et al., 1998).
Processing of SAR data closely mimics the processing steps described for optical data, with the following distinctions:

All Sentinel-1 SLC (Level 1.1) of TOPS IW mode data are pre-processed using the NASA/JPL's InSAR Scientific

Computing Environment Version 2 (ISCE2) software (https://github.com/isce-framework/isce2) prior to dense offset-
tracking, where the two SLC images are precisely co-registered using the satellite orbit geometry. All SAR images are
preprocessed using a 21x21 Wallis operator to normalize for local variability in radar backscatter caused by topography,
followed by a 32-bit floating point to 8-bit integer data compression to save space and improve efficiency. Pre-processed
same-path-frame pairs of images are searched for matching features by finding local normalized cross-correlation (NCC)
maxima at sub-pixel resolution by oversampling the correlation surface by a search-chip-size-dependent factor. Correlation
surface oversample values of 32, 64, 128 and 128 are used for chip sizes of 240 m, 480 m, 960 m and 1920 m, respectively,
using a Gaussian kernel (Lei et al., 2022). The search-chip-size-dependent factor is used to match the oversampling ratio
with maximum achievable precision from the data.

### 3.1.4 Velocity uncertainty in Level 2 image pairs

Sources of uncertainty in ITS_LIVE Level 2 velocity data products are related to the accuracy of the geolocation

that can be obtained for an image pair, and the quality of the correlation peak for a given sub-image pixel match. The
observed initial offset error, assessed as the uncorrected offset to stable surfaces (rock or slow moving ice), averages under
half a pixel for both Landsat 8 and Sentinel 2A-2B, less for Sentinel 1A-1B, but can be as large as a full pixel (15 m for
Landsat 8). Geolocation offsets are corrected by adjusting scene-pair velocities to known stable surfaces such as rock or
slow-moving ice, and after correction, displacement accuracy is better than a tenth of a pixel (Lei et al., 2021). Correlation-
related errors are also on the order of less than a pixel where correlation peaks are distinct. With restrictive masking of
weakly correlated offset matches, remaining offsets over stationary targets have conservative root-mean-square errors of
0.1 pixels, which translate to conservative estimates of individual velocity errors of ~1 m/day for a pair of Landsat images
separated by 16 days, or 0.12 m/day error for a pair of Landsat images separated by 96 days, in agreement with similar data
products (Mouginot et al., 2017). Offset errors are significantly smaller with larger search chip sizes, and, as each search
chip used for offset determination contains 50% overlap with adjacent chips, the surrounding offsets can be averaged with
the  error decreasing as:
$$\underline{v}_{x/y\,error} \cong \frac{mean(v_{x/y\,error})}{\left(\frac{n}{4}\right)^2}$$


where n is the number of offsets averaged. To correct for geolocation errors, component velocities $v_x$ and $v_y$ are tied to a
"stable" surface wherein the median of each velocity component is set to zero over rock surfaces and set to the median



reference velocity over slow-moving areas (ice movement of less than 15 m yr$^{-1}$) of Greenland and Antarctica. If an image
pair does not intersect a stable surface, an alternative error metric included with almost equivalent performance that uses
the area of the slowest 25% of the reference velocity. After geolocation corrections are applied, velocity uncertainty in each
component direction is calculated as the root-mean-square of measured velocities over the stable reference surface. An
additional error metric **v_error** is calculated as:

$$v_{error} = \sqrt{\left(vx_{error} * \frac{vx}{v}\right)^2 + \left(vy_{error} * \frac{vy}{v}\right)^2}$$

SAR orbit and viewing geometry between same-path-frame image pairs are highly stable, so the geolocation error
is very small for an image pair consisting of the same satellite (both images from Sentinel-1A or both from Sentinel-1B).
However, the inter-satellite (Sentinel-1A/B) image pairs suffer from subswath-dependent and full swath-dependent
geolocation errors due to systematic issues. To correct for subswath-dependent geolocation error, 11 Sentinel-1A/B image
pairs were characterized over the interior of Greenland with slow-moving ice surface, and the inter-swath range/azimuth
pixel offset bias estimates are then used as a static correction of the subswath-dependent geolocation error in each ITS_LIVE
Sentinel-1 image pair product (Lei et al., 2022).

### 3.2 Data cubes

After generating Level 2 image pair data, a cloud optimised data cube product is generated internally to collect
Level 2 data for composting and mosaicking. In this process, each glacierized region is subdivided into 100 km by 100 km
tiles, and Level 2 data within each tile are stored as layers in a Zarr file.

### 3.3 Level 3 composites and mosaics

Individual annual and climatological composites are created for each 100 km by 100 km data cube. As a first step,
Level 2 optical and SAR data undergo numerous quality controls to account for issues related to geolocation errors, sensor-
specific performance, and feature-locking, all of which are described in Appendix A. Filtered Level 2 optical and SAR
image pair velocities are combined to form annual composites for data cube using an error-weighted least-squares approach
that simultaneously solves for the mean annual velocity and a sinusoid that characterizes the climatological average seasonal
cycle (Greene et al., 2020). In this approach, total displacement measured between the acquisition times of each image pair
is fit to an amplitude and phase of a sinusoid, and a constant value corresponding to each year. Displacement coefficients
are then converted to velocity values to obtain annual mean velocity values **vx**, **vy**, and **v**. The result is a mathematical best-
fit characterization of typical seasonal velocity variability that accounts for total displacement over long polar winters when
optical data are often unavailable, and an annual mean velocity value that is unbiased by the timing of image acquisitions





throughout the year. In this process, outlier observations determined by a median absolute deviation filter are discarded after
an initial fit to all data, then the fit is repeated with outliers removed.

In addition to annual velocity values, overall summary climatology composites are generated for each region as

Level 3 data files containing **0000** in place of a year in the filename. Summary velocities are calculated using a least-squares
fit applied to image-pair data with a mid date between January 1, 2014 and January 1, 2023. Seasonal components (**vx_amp**,
**vy_amp**, **v_amp**, **vx_phase**, **vy_phase**, **v_phase**) are determined directly from the least squares fit. The mean velocity and
velocity trend are determined from an error-weighted linear fit to the annual values. We then solve the fit velocity for an
arbitrary date of July 2, 2018 to create a consistent map of velocity with minimal spatial variation that might otherwise be
caused by a simple mean of the data available in each grid cell. The intention here is to create a best snapshot of flow that
can be used in mass-conserving divergence or gradient calculations, with a consistent effective timestamp across all pixels.
The slope of the linear fit to annual values is also provided as velocity trends **dvx_dt**, **dvy_dt**, and **dv_dt** in the summary
mosaic file. Note that the date range used for climatology calculations will be updated as the record lengthens. It is
recommended that users inspect product metadata to confirm the dates of the input data.

Regional mosaics are then created by re-projecting 100 km by 100 km composites into a common projection and

merging. When reprojecting that data, care is taken to rotate and scale the velocity components to be consistent with the
target map projection. Overlapping composites are weighted by data counts. Count is taken as the maximum count of
overlapping composites. ITS_LIVE produces mosaics for the 16 regions shown in Figure 2 that cover the majority of
glacierized area. For areas that fall outside of these 16 regions, users can work directly with the unmerged composites.






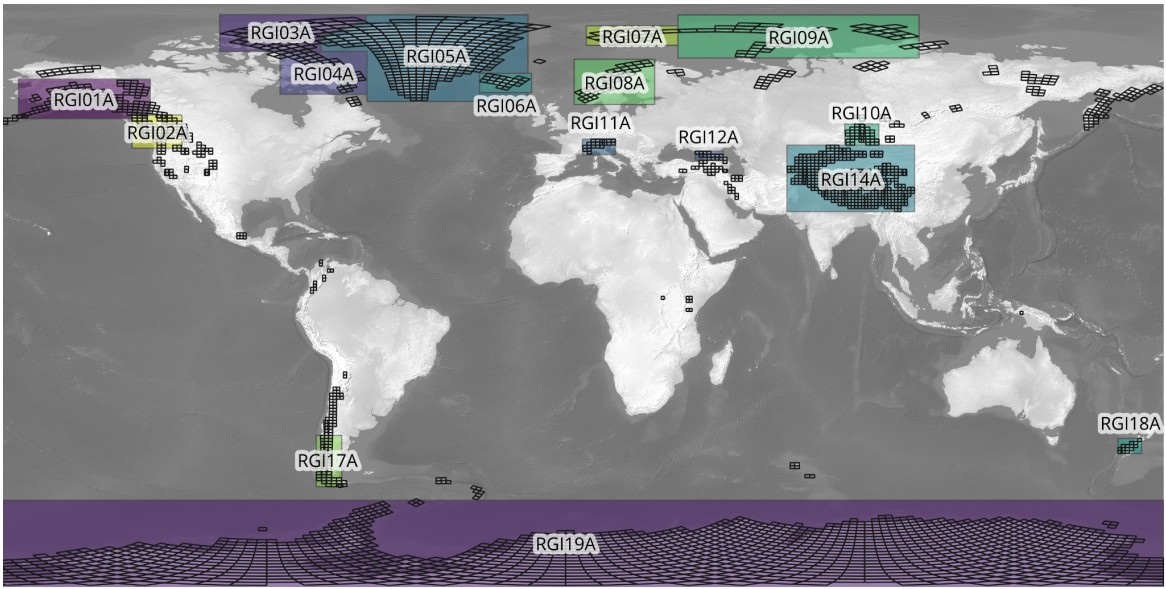


**Figure 2. Shaded rectangles show the coverage of the 16 ITS_LIVE mosaic regions. Regions follow a similar naming convention as the Randolph Glacier Inventory Version 6. All 3086 100 km by 100 km ITS_LIVE data cubes are shown with black outlines.**

### 3.3.1 Error estimates and quality metrics

Annual velocity mosaics contain a **count** variable indicating the number of image pairs that at least partially contribute to the error-weighted least-squares fit for that year. Annual mosaics also contain estimates of **vx_error**, **vy_error**, and **v_error**, which are error-weighted means of error estimates of all contributing Level 2 data for each year.

Climatology mosaics containing **0000** in the filenames include a **count** variable indicating the total number of image pairs used to estimate the climatology velocities, trends, and seasonal variability and an **outlier_percent** indicating the percentage of Level 2 data excluded from the Level 3 climatology fit. Formal errors from Level 2 products are propagated through to the annual mosaics but can produce an overly optimistic estimate of product errors, so we adopt a more conservative approach and calculate the errors as the standard error of the mean. This is calculated by taking the root-sum-of-squares of the residuals to the least squares fit and dividing by the number of observations. Annual errors are then propagated to determine mean flow mosaics errors. Estimates of **vx_amp_error**, **vy_amp_error**, and **v_amp_error** describe the overall mismatch of velocity observations to the seasonal fit. Uncertainty of seasonal phase values cannot be estimated formally, but is expected to be accurate within a few days or weeks where amplitudes are significant and hundreds or more image pairs contribute to the fit (Greene et al., 2020).





## 4 Data counts and global statistics


ITS_LIVE version 2.0 global velocity mosaics describe the flow of 14,190,690 km$^2$ of Earth's land ice, covering
every glacier larger than 5 km$^2$ north of 83°S. The data reveal diverse landscapes of glacier flow and complex responses to
an evolving climate. The summary mosaics show that the world's fastest land ice is in Greenland, where the central trunk
of Sermeq Kujalleq (Jakobshavn Glacier) exceeds 10,000 m yr$^{-1}$. The fastest grounded ice in Antarctica flows from Pine
Island Glacier into the Amundsen Sea at more than 4000 m yr$^{-1}$, and the fastest glacier outside the great ice sheets is Hubbard
Glacier in Alaska, where average speeds exceed 3000 m yr$^{-1}$. Annual mosaics show Alaska's Columbia Glacier exhibits
velocities  comparable to Hubbard Glacier near its terminus, but its top speed is not accurately reflected in the climatology
due to rapid retreat that has been ongoing there since the 1980s. Elsewhere in the Arctic, Storisstraumen Glacier (Basin 3)
in Svalbard has averaged nearly 3000 m yr$^{-1}$ while slowing steadily at a rate of 200 m yr$^{-2}$ over the past decade. Glaciers in
mid- to low- latitudes are generally characterized by slow velocities. However, areas of fast flow are observed for the largest
glaciers (e.g. Pio XI glacier, Patagonian icefield reaches 3000 m/yr, Fedchenko glacier, Pamir, reaches 800 m/yr), at
localized icefalls (Khumbu icefall, Nepal, up to 400 m/yr; Bossons icefall, French Alps, up to 500 m/yr) or during glacier
surges, which are particularly prevalent in the Pamir and Karakoram (Khurdopin glacier, Karakoram, peaked above 3500
m/yr in May 2017).
Level 3 summary mosaics of ITS_LIVE version 2.0 confirm that glaciers around Greenland have accelerated over
the past decade, and Antarctica's most significant dynamic changes are concentrated in the Amundsen Sea Embayment,
most notably at Pine Island and Thwaites glaciers. Although velocity variability is seen in every region of the globe, glaciers
have not responded uniformly to recent climate change. In the data, we do not see any definitive global bias toward glacier
acceleration or deceleration since 2014, but we do see subtle regional trends, and a diversity of behaviours within each
region. The largest magnitudes of **dv_dt** in Alaska are driven by surges, and the sign of these linear trends are closely linked
to the timing of surge activity. For example, the velocities of Seward, Steller, and  Lowell glaciers all trended upward in the
past decade due to recent surge events, while the velocities of Fisher, Walsh, and Klutan trended downward following surge
events that initiated in 2015 and 2016. Similarly, several glaciers in Svalbard and the Canadian Arctic show decadal velocity
trends that can be attributed to timing of surge events.
Globally, the highest concentration of high-amplitude seasonal variability is observed in Alaska, where velocities
tend to peak in spring or summer. In contrast, some glacier velocities in High Mountain Asia peak in spring, while others
peak in fall. We confirm previous reports of seasonally variable discharge in Greenland that peaks in summer (King et al.,
2018), but find little evidence of seasonal variability around Antarctica, particularly on grounded ice.
In total, more than 36,000,000 image pairs currently contribute to ITS_LIVE version 2.0, beginning with the 1982
launch of Landsat 4. The record is somewhat sparse globally until the 2013 launch of Landsat 8 (Figure 3), which was
followed by yearly launches of Sentinel satellites through 2017 and the launch of Landsat 9 in 2021. Now, almost every





grid cell in the world is captured by multiple satellites each year, velocity is directly measured throughout long polar nights
with the Sentinel 1 satellites, and some locations are characterized by as many as 100,000 velocity estimates per year. The
sheer volume of observations now available suggest that glaciology is no longer a field held back by data starvation.




**Figure 3: Time series of the number of Level 2 image pairs contributing to each grid cell of each Level 3 annual velocity mosaic of Greenland. Satellite names appear in the first year they contribute to an annual mosaic in ITS_LIVE version 2.0.**


**4.1 Limitations and uncertainties**

Although Level 2 image pairs can be processed within minutes of satellite image availability, some appreciable
lag is necessary before Level 3 mosaics can be generated. Because we process image pairs separated by up to 546 days,
some observations covering December 2023, for example, will not be available until their paired images are acquired in
June 2025. To ensure that all available data are included in annual mosaics, ITS_LIVE generates Level 3 mosaics in
dedicated campaigns after all contributing images are acquired.
Level 2 image pairs are most accurate where rock or other stable surfaces are present within the satellite image
frame for georeferencing. Data users should be aware that feature tracking directly measures displacements, and precision
is limited by image pixel size and image quality. Velocity is calculated as displacement over time, so errors in velocity can
be mitigated by increasing the time between images (*dt*), but long separation times between images come at the cost of
temporal resolution and can allow the surface to change or lose its distinguishing features between image acquisitions. As
described above, at some locations near ice edges, ice falls, or bends in glacier flow, surface patterns may be replicated with



enough similarity that long values of *dt* confuse the feature-tracking algorithm by allowing it to skip or lock onto the
incorrect pattern cycle. A filter is in place to identify and discard velocity values that likely correspond to skipping or locking
before Level 3 mosaics are calculated (Section 3.3 and Appendix A), but users of Level 2 data should be aware of the
potential benefits and risks of using image pairs with short versus long *dt* values.
Level 3 velocity uncertainty reduces where an abundance of Level 2 observations are available, so mosaics
generally have lower uncertainty values toward the poles, where satellite orbit patterns converge and many images overlap.
Low data counts can result from sparse orbits, persistent cloud cover that obscures optical images, or high accumulation
rates that create featureless or frequently changing surfaces that cannot be tracked. For example, data counts are particularly
low along the high peaks of the Southern Andes, where cloud cover is common and accumulation rates are high. Whereas
the median data count among land-ice grid cells is 5407 in the summary mosaic of Region 5 (Greenland), the median value
is only 881 in Region 17 (South America), and some high elevation locations in this region have nearly no valid data at all.
Metrics of seasonal variability are most accurate where several hundred or more image pairs contribute to the
sinusoid fit (Greene et al., 2020). This condition is met in most locations around the world, but where data counts are lower,
the least-squares fit becomes especially sensitive to extreme velocity values and **v_amp** may be larger than the true
amplitude of seasonal variability. We note that the sinusoid fit to the seasonal cycle is a best-fit model that describes only
the fundamental mode of seasonal variability and does not account for higher-order acceleration/deceleration or changes in
seasonal behaviour from one year to the next. The timing of the maximum velocity in a sinusoid fit may not align with
ephemeral spikes in velocity, and although filters are in place to remove outliers before performing the final fit, discrete
events such as glacier surges can in some cases contaminate the overall seasonal characterization. We recommend exploring
the complete time series of Level 2 data at any given location to interpret the summary mosaic metrics of seasonal variability.
Similarly, interpretation of **dv_dt** velocity trends since 2014 may warrant inspection of the complete time series to determine
any potential influence of surge type behaviour that is nonlinear by nature and cannot be accurately characterized by a linear
fit.
The update of ITS_LIVE velocity data from version 1.0 to version 2.0 includes a change in reporting of velocity
values. Whereas version 1.0 included a correction for map distortion, velocities in version 2.0 are now reported in map units
of the projection in which the data are published. Projection distortion can be on the order of a few percent in some locations
and should be corrected when comparing to in situ observations (e.g., GPS), but the change was made for consistency within
the data product, to allow calculation of flow lines from the velocity components in map projected coordinates, and to allow
flux estimates that no longer require the flux-gate cross-section to be corrected for map distortion.





## 5 Data access and tools

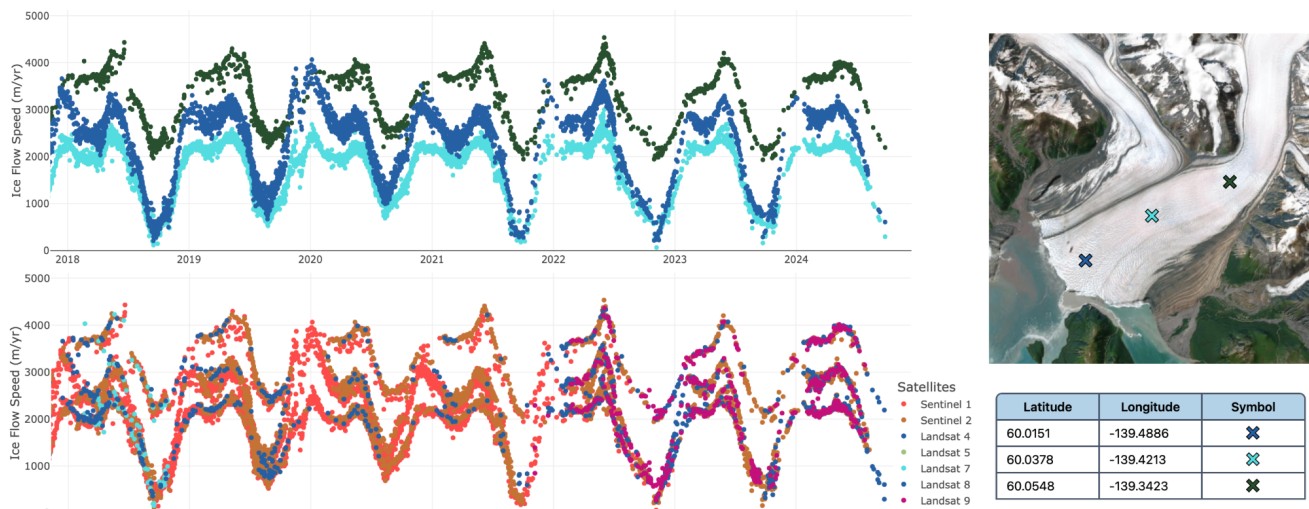

**Figure 4: An example velocity record for lower Hubbard Glacier, Alaska. Upper left: ice flow speed for three points shown in the map on the right. Lower left: The same data, color-coded by satellite. Sentinel-1 (red) adds significant detail and sampling density in this very cloudy coastal setting, and provides speeds through the winter. Basemap from Earthstar Geographics (TerraColor NextGen) imagery. An interactive version of this figure can be accessed at [https://its-live.jpl.nasa.gov/app/index.html?z=11&lat=60.0151&lon=-139.4886&lat=60.0378&lon=-139.4213&lat=60.0548&lon=-139.3423&int=1&int=100&x=2017-11-10&x=2024-12-12&y=-31&y=5115].**

ITS_LIVE version 2.0 velocity data are hosted on Amazon AWS as part of the AWS Open Data Sponsorship Program and served through NASA's National Snow and Ice Data Center Distributed Active Archive Center (NSIDC DAAC). The data can be accessed in several ways according to user needs and preferences.

Level 2 image pairs and Level 3 mosaics can be accessed through the NSIDC using the ITS_LIVE app at https://nsidc.org/apps/itslive/. In this app, users can pan and zoom an interactive map to find Level 3 annual and summary mosaics for each region or build search queries to find and download Level 2 image pair granules as NetCDF files. A suite of other Python tools and notebooks are available at https://github.com/nasa-jpl/its_live and similar utilities for data download and analysis are available for Julia and MATLAB users, links to which are provided on the main ITS_LIVE website (https://its-live.jpl.nasa.gov) along with other helpful information.

The Level 2 velocity pipeline generates a large amount of data, such that any single point on Earth may be covered by 100,000 image pairs or more. To eliminate the need for users to download or open such a large number of granules, ITS_LIVE also provides all Level 2 data in 3,086 100 km by 100 km cloud optimized Zarr data cubes that are structured



for rapid time-series access. The data cubes retain all fields and metadata from the original Level 2 data, and are hosted on
public AWS S3 cloud storage, allowing users to write generic workflows that do not require downloading tens or hundreds
of thousands of individual Level 2 NetCDF files. ITS_LIVE also hosts a geojson catalogue of all of the data cubes on public
S3 cloud storage. Utilities for working with Zarr data cubes can be found at https://github.com/nasa-jpl/its_live.

The adoption of cloud-hosted, cloud-optimized Zarr data cubes has enabled the creation of a serverless web app
(Figure 4) that allows users to interactively explore glacier velocity time series at any location on Earth. The interface allows
users to enter geographic coordinates or select points on a map to instantly plot the full record of velocity data at specified
locations. Users can then share a hyperlink to the same map with collaborators or students, for frictionless workflows and
lesson plans that are open and replicable. The map interface can be accessed at https://its-live.jpl.nasa.gov/app/index.html.

**6 Conclusions**

One of ITS_LIVE's goals is to provide as complete a record of ice flow as practical from online imagery collections -
an observational record of glacier flow that is global in scope, processed with open source tools in a consistent way, that
can be extended with new data as it is acquired. ITS_LIVE has now processed more than 36 million satellite image pairs,
spanning the globe and covering four decades and counting. The data are free and easily available in multiple file formats,
can be accessed locally or in the cloud, and open-source tools are available in multiple computing languages to help users
access, analyse, and understand the data. The release of version 2.0 preserves traditional NetCDF granule access while also
supporting cloud-native Zarr access for modern big-data machine learning applications. Level 2 image pairs are available
for process studies that require high resolution reconstructions of dynamic time series; Level 3 mosaics can easily be
employed to estimate ice-sheet mass discharge and sea level contributions; and together, the ITS_LIVE products will enable
precise ice-sheet and glacier model calibration and validation for improved projections of future changes in Earth's climate
system. Version 2.0 velocity products compliment additional ITS_LIVE geophysical data products of surface elevation
(Nilsson et al., 2022, 2023; Nilsson and Gardner, 2024), ice-shelf basal melt (Paolo et al., 2024, 2023), and ice-sheet extent
(Greene, 2024; Greene et al., 2022, 2024), which together aim to characterize changes in the world's ice in every dimension,
in usable, interoperable formats.

Version 2.0 of ITS_LIVE velocity products have been processed at 120 m resolution globally, which is an
improvement over the 240 m resolution of the version 1.0 products. Future releases of ITS_LIVE velocity data products are
expected to include data from the Sentinel-1C and NASA-ISRO Synthetic Aperture Radar (NISAR). ITS_LIVE is also
assessing ways to fill in the historical archive with data from the Advanced Spaceborne Thermal Emission and Reflection
Radiometer (ASTER) instrument that was launched aboard the Terra satellite in 1999 and RADARSAT-1 that was launched
in 1995 and decommissioned in 2013.



429   The dense spatiotemporal coverage of the ITS_LIVE version 2.0 and future releases will help scientists discover

430  previously unknown patterns of glacier flow and the mechanisms that cause and control them. In the data, users will find

431  glaciers surging, shear margins migrating, kinematic waves propagating up and down glaciers, dynamic responses to calving

432  events and ice-shelf thinning, and speedups and slowdowns driven by seasonal changes in basal hydrology. A world of new

433  insights now reside in ITS_LIVE version 2.0, and are waiting to be discovered.





**Appendix A**

The following details the processing steps used to generate composites from the multi-sensor data cubes (one composite per data cube):

    A.  Add systematic errors to image pair component velocity errors, based on level of co-registration as indicated by the "**stable_shift_flag**" attribute that is included with the product. The **stable_shift_flag** is a flag for tracking the velocity bias correction: 0 = no correction; 1 = correction from overlapping stable surface mask (stationary or slow-flowing surfaces with velocity < 15 meter/year)(top priority); 2 = correction from slowest 25% of overlapping velocities (second priority). A random error for each image-pair is provided with each granule and is calculated as the standard deviation between the reference and the measured component velocities over the co-registration surface (i.e. stable or slowest 25%). A default random error is assigned when stable_shift_flag = 0. We add an additional systematic error to the random errors as listed in Table 1A.

**Table 1A: Systematic error added to v[x/y] _error as a function of stable_shift_flag**

| stable_shift_flag | vx_error | vy_error |
|---|---|---|
| 0 | vx_error_random + 100 m/yr | vy_error_random + 100 m/yr |
| 1 | vx_error_random + 5 m/yr | vy_error_random + 5 m/yr |
| 2 | vx_error_random + 20 m/yr | vy_error_random + 20 m/yr |

    B.  Over ice sheets it was found that image-pairs that were co-registered to limited areas of "stable" surfaces could contain unrealistically small velocity errors. These unrealistically small errors cause issues with the error-weighted composite generation. To correct for this, in regions RGI05A (Greenland) and RGI19A (Antarctica), we replaced **vx_error** and **vy_error** with **vx_error_slow** and **vy_error_slow**, respectively.

    C.  Apply a StableShiftFilter. This routine discards low-quality image-pair data that have absolute vx/vy **stable_shift** values that exceed per each mission group thresholds (thresholds determined from histograms of **stable_shift** values for each sensor) that are listed in Table 2A.

**Table 2A: Mission specific StableShiftFilter thresholds**

| Mission | StableShiftFilter threshold |
|---|---|
| Landsat 4/5 | Infinite |
| Landsat 7 | Infinite |





| Landsat 8/9 | 61.6 m |
|---|---|
| Sentinel 1 | 1.1 m |
| Sentinel 2 | 28.5 m |


If an image-pair exceeds the mission specific threshold in StableShiftFilter (i.e. very large vy_stable_shifts
have been applied) then the following actions were taken: If **stable_shift_flag == 1** then the image-pair
was excluded and if **stable_shift_flag == 2** then the stable shift was removed from the velocity field (i.e.
added back to vx/vy/vr/va).
D. ITS_LIVE velocities are produced by finding correlated features between two image chips, a process
referred to as feature tracking. In some locations, feature tracking can be susceptible to surface "skipping"
or "locking", where instead of tracking the surface features that are the intended targets, the correlation
incorrectly locks onto features that have shapes that are similar to the intended target features (Figure A1).
The problem is caused by high-frequency radiometric features that are not removed by the high-pass filter,
and are stationary because of topography, surface water, curved flow lines (constrains both x and y),
crevasse chains or some combination of all. Radar speckle tracking will also suffer from "skipping" where
high frequency stationary features exist in the amplitude image (ice falls, curved flow lines, surface water).
The degree of locking/skipping depends on the surface features, sensor characteristics (spatial resolution,
radiometric resolution), the high-pass preprocessing filter and the search chip size. The three places where
skipping/locking is most prevalent is near ice edges, ice falls and flow bends.





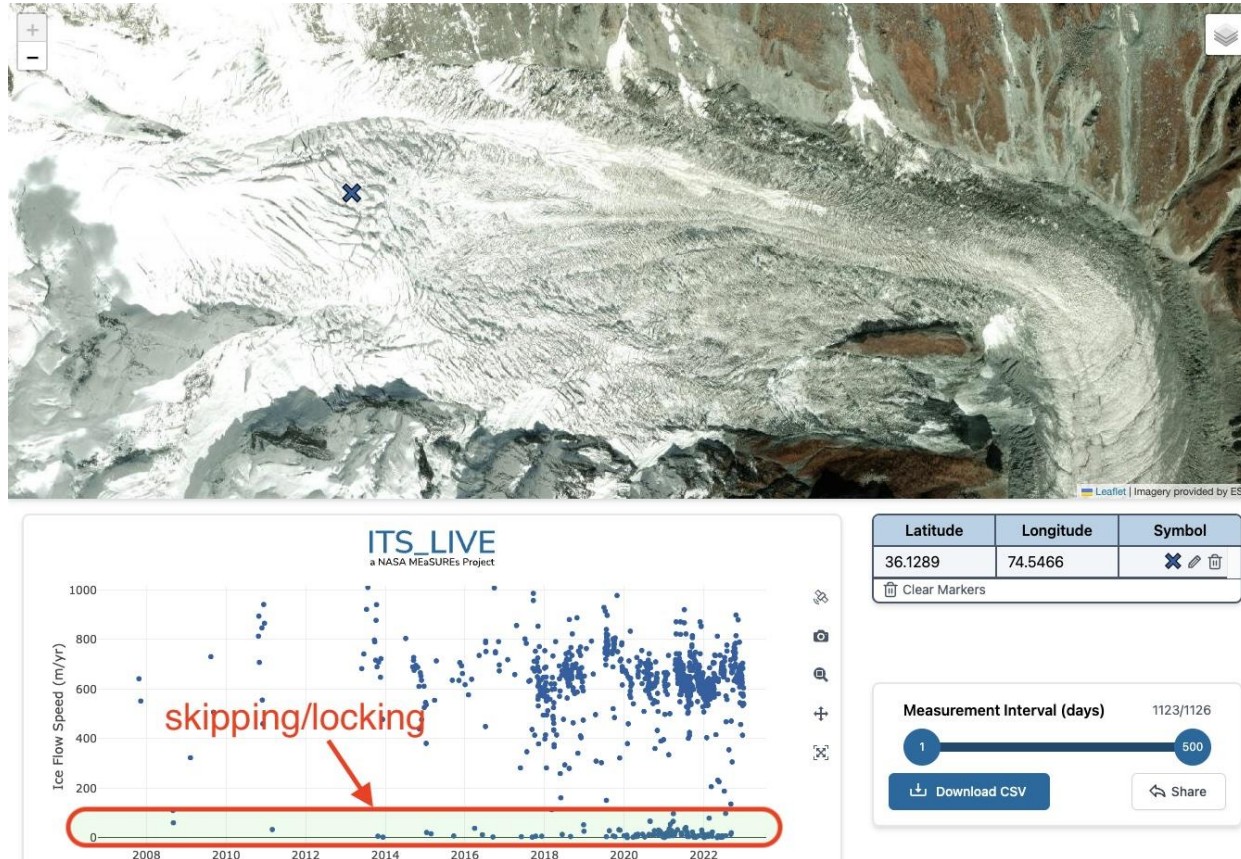

**Figure A1. Example of surface skipping/locking feature tracking matches that result in periodic near-zero velocities. Retrieved flow speeds are shown for blue "x" that is located at the edge of an ice fall. Flow speeds are generally in the range 300-1000 m/yr. The clustering of flow speeds near 0 m yr⁻¹ are erroneous and result from surface skipping/locking. Basemap from Maxar (Vivid) imagery captured on Oct 15, 2022. Image generated from ITS_LIVE app for data cube exploration: https://its-live.jpl.nasa.gov/app/index.html?z=15&lat=36.1289&lon=74.5466&x=1987-04-16&x=2022-12-27&y=-131&y=1401&int=1&int=500**

Sensors that have a lower spatial and/or radiometric resolution and image-pairs that are acquired further apart in time are most prone to surface skipping/locking. We apply the SensorExcludeFilter to identify and remove surface skipping/locking errors as follows:



    a.   Load landice_2km_inbuffer mask for the data cube being processed. landice_2km_inbuffer is a binary mask that defines the extent of glacierized areas after applying a 2 km inward buffer from the glacier edge.

    b.   Define sensor groups: Landsat 4/5, Landsat 7, Landsat 8/9, Sentinel 2, Sentinel 1

    c.   Exclude any data that has a velocity magnitude > 20,000 m per year from further analysis.

    d.   Apply SensorExcludeFilter for locations that are within 2 km of the ice edged: landice_2km_inbuffer mask == 0. The following steps are taken when applying the SensorExcludeFilter:

        i.   Sentinel-2 image-pair data is used as the reference sensor group as it is least prone to skipping/locking errors. If there are no Sentinel-2 granules for a given location then the SensorExcludeFilter is not applied.

        ii.   Only include image-pair data with time separation less than or equal 64 days. This is done as image-pair data with longer time separations are more prone to skipping/locking errors.

        iii.   For each sensor group we compute mean vx and mean vy and then calculate the unit vector. All image-pair vx and vy are then projected onto their corresponding sensor group unit vector.

        iv.   Projected velocities are binned into 1/5 of a year bins spanning the time range of the Sentinel 2 data. For bins with more than three values the following statistics are calculated: mean projected velocity, standard deviation and count. If the reference sensor (Sentinel-2) has no bins with more than three values then the SensorExcludeFilter is not applied.

        v.   For each non-reference sensor group we identify bins that are valid for both the sensor group and the reference sensor. If there are fewer than three co-valid bins then we do not apply the SensorExcludeFilter to that sensor group. If there are more than three co-valid bins we compute standard error between the co-valid projected velocities. If the mean of the co-valid sensor group values is 3 times the standard error below the mean of the reference sensor, then the sensor group is excluded from composites calculation at that location. Here, excluded values are likely experiencing significant skipping/locking errors and therefore should not be included in the composites.

   E.   Next, we apply the MaxDtFilter that determines the maximum image-pair time separation that should be included in the composite creation. This is done to minimize skipping/locking errors that are more



prevalent with increasing image-pair time separations. A maximum image-pair time separation (dt_max) is determined for each sensor and each location as follows:

    a. Calculate the median composite velocities for all image-pair data with time separation less than or equal 16 days. Each point must have at least 50 valid values. If a location has fewer than 50 valid values, the time separation threshold is progressively increased from 16 to 32 to 64 to 128 to 256 to infinity until at least 50 valid values are identified. If this condition is never met then the location is set to no-data in the composite creation. Where the condition is met, we calculate the median velocity magnitude and unit flow vector from the median composite velocities.

    b. If the median velocity magnitude is less than or equal to 50 m/yr then MaxDtFilter is not applied.

    c. Project all image-pair velocities to the median unit flow vector

    d. Bin projected velocities by image pair time separation into bins with edges 0, 16, 32, 64, 128, 256, and infinity days

    e. For each bin calculate the median, count, and the median absolute deviation from the median times 1.4826 to make it a consistent robust estimator to the standard error (NMAD).

    f. Compute minimum and maximum projected velocity bounds for each bin based on median ± (NMAD * 0.67)

    g. Identify a reference bin as the first bin with 50 or more velocities moving from bin 0 to 16 days through to 256 to infinity days. If no such bin exists, the reference bin is set to the first bin with two or more velocities. If no such bin exists, MaxDtFilter is not applied.

    h. Find the first bin, moving from bin 0 to 16 days through to 256 to infinity days, that does not have overlapping bounds with the reference bin. Set maximum allowed time separation (dt_max) equal to the lower bound of the identified bin. If all bins overlap then dt_max is not applied.

    i. For composite creation, only include data for which the image-pair time separation is less than or equal to dt_max.

F. Determine annual and climatological glacier velocities for each 120 m pixel location following Greene et al. (2020):

    a. Apply a 15 point moving window filter to all input velocity data.

    b. Create a matrix M of coefficients that define the percentage of each year spanned by each image-pair. The matrix M is used in the least-squares calculation to obtain a mean annual velocity corresponding to each year.

    c. Calculate the least squares weighting for each value as 1 divided by the square of the displacement error (velocity error time dt).





d. Determine annual composite outputs as the optimal fit of all valid data in an error weighted least squares sense.

e. For climatological composites we do the same least squares fit but only include image-pair data with a mid date between January 1, 2014 and January 1, 2023. Mean velocity and velocity trend is determined from an error weighted linear fit to the annual data (time-intercept of January 1, 2018).

G. In areas distant from the ice edge (landice_2km_inbuff == 1) and with low radiometric contrast, Sentinel 2 image-pair velocities can contain high noise due to image processing artifacts. These artifacts can introduce significant noise into the composite creation. To mitigate this, we apply a S2Filter as follows:

a. Recompute the annual and climatological composite outputs excluding all Sentinel 2 data.

b. If the original seasonal amplitude is twice as large as the recomputed seasonal amplitude, and the difference in the seasonal amplitudes is greater than 2 m per yr, then exclude Sentinel 2 data, at this location, from the composites.

H. If annual velocity magnitude is greater than 20,000 m per year then all data for that year is excluded. If the seasonal amplitude is greater than 10,000 m per year then all data for that point is excluded.





I.

## Code availability

All code created for the ITS_LIVE project is open sourced:

- The autoRIFT feature tracking software is located at https://github.com/nasa-jpl/autoRIFT.
- The Hyp3 ITS_LIVE image monitoring software is located at https://github.com/ASFHyP3/its-live-monitoring.
- The Hyp3 autoRIFT deployment software is located at: https://github.com/ASFHyP3/hyp3-autorift.
- The serverless web application is located at: https://github.com/nasa-jpl/itslive-web.
- Python tools for working with the ITS_LIVE data are located at: https://github.com/nasa-jpl/itslive-py

## Data availability

All ITS_LIVE products are freely accessible and can be found at the following locations:

1. The NASA National Snow and Ice Data Center Distributed Active Archive Center (NSIDC DAAC): https://nsidc.org/data.
2. The Amazon Web Services through support of the Open Data Sponsorship Program: https://registry.opendata.aws/its-live-data.

## Author contributions

ASG, MF and TS conceived of the ITS_LIVE project. ASG wrote its underlying autoRIFT software and the composite algorithms. JHK built the HyP3 autoRIFT plugin, deployed and managed HyP3 ITS_LIVE, and built the ITS_LIVE monitoring stack. MF developed the image pair-picking strategy for the optical missions, JHK developed the stragey for Sentinel-1managed the processing, and MF and JHK managed the image processing campaigns. CAG contributed to algorithm development, data product testing, and coordinated the writing of the manuscript. YL translated autoRIFT into Python and added support for Sentinel-1 processing. ML productionized the composite code and constructed the data cubes. LL wrote the NSIDC ITS_LIVE data discovery application. JF wrote the JAVAscript web application. Python tools and notebooks were largely written by LL and MF. All authors contributed to the writing of this manuscript.

## Competing interests

The authors declare no competing interests.



**Acknowledgments**
The authors were supported by the ITS_LIVE project awarded through the NASA MEaSUREs program. A portion of this
research was carried out at the Jet Propulsion Laboratory, California Institute of Technology, under a contract with the
National Aeronautics and Space Administration (80NM0018D0004). The authors would like to thank the ITS_LIVE user
community for their helpful feedback, the ASF Tools Team for their data processing advice and support, the Pangeo
community, the ESA for processing the Copernicus Sentinel data, the U.S. Geological Survey for Landsat data, and the
NASA MEaSUREs program for funding ITS_LIVE. We are grateful to the Amazon Web Services (AWS) Open Data
Sponsorship Program that covers the cost of storing the ITS_LIVE archive on AWS.

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
