# Peer review of "ITS LIVE global glacier velocity data in near real time"

_EGUsphere, 2025_

## Author Response (AR1)

Dear Dr. Enderlin,

Thank you for your time, kind words and thoughtful review. All your suggestions were great. We've addressed all as follows:

Minor Comments:

- lines 35-37: I recommend rephrasing this sentence because it currently anthropomorphizes glaciers a bit too much, albeit unintentionally.

  *We agree and have revised the starting sentences of the introduction to be more focused.*

- line 84: I found this to be the most confusing statement in the processing description: "Each new image may pair with up to 35 previous images and create 35 new velocity granules". Later in the main text and in the appendix you describe velocity pairing in more detail and the focus is already on time separation, not the number of images. Why do you limit the search based on the number of images here? Since you look for up to 35 images, does that really define a maximum time frame (at least for the same path-row)? Please clarify.

  *Yes, this is confusing. The "35 previous images" is simply a function of the allowed time separation between images when forming an image pair. We've removed this sentence as it is redundant with later text and is out of place in the "monitoring" section.*

- lines 150-151: I do not see how it is possible the generate velocities on a uniform grid without any resampling or interpolation when you are bringing together such different datasets. A detailed explanation of geogrid is outside the scope of this paper but it would be helpful to clarify this statement about geocoding of the autoRIFT outputs so that the process is more transparent for people who use this data product.

  *We agree that our description is not easy to intuit. To address this we've replaced the url to the Geogrid repo with the citation to the paper in which the algorithm is described in detail. We've also added an additional sentence that hopefully adds clarity: "This is achieved by centring search chips on a predefined grid then mapping these locations to native image coordinates, accounting for rotations and distortions between mappings".*

- lines 163-171: A small table describing each dataset would be really helpful. The time period of data, band name and wavelength or frequency as appropriate, and spatial resolution are all really helpful parameters to know.

  *Great idea. We've added a new table with this information.*

- line 190: What source are used for reference velocities? Also, you mention that a DEM can be used in autoRIFT when describing HyP3 autoRIFT. Do you use a DEM? If so, from what source(s)? Does the DEM also provide geographic constraints on the search? Please explain.

  *We've added information on the reference velocity: "Our reference velocity is derived a synthesis of Version 1 MEaSUREs ITS_LIVE Regional Glacier and Ice Sheet Surface Velocities (Gardner et al. 2022), MEaSUREs Version 1 of the Multi-year Greenland Ice Sheet Velocity Mosaic (Joughin et al. 2016), and Version 1 MEaSUREs Phase-Based Antarctica Ice Velocity Map (Mouginot et al., 2019)."*

  *And on the DEM used: "We use the Global Copernicus GLO-30 Digital Elevation Model in our SAR processing."*

- line 202: I'd move this sentence to the start of the next paragraph since that paragraph focuses on the differences between optical feature tracking and speckle tracking.

  *Good catch, thanks.*

- lines 217-218: Landsat 8 is mentioned twice and one instance has to be a typo. *Fixed*

- line 246: Typo "compositing"

  *Fixed*

Sincerely,

Alex and co-authors.

Dear Reviewer,

Thank you kindly for taking the time to review our manuscript and for the kind words. We've addressed all your comments in the revised manuscript, which we detail here:

Comments to the Authors:

1. The manuscript notes that ITS_LIVE uses both optical and SAR data, but it would be helpful to explain more clearly the advantages of incorporating SAR. Is it for higher temporal/spatial coverage, better performance in cloudy regions, or increased measurement accuracy? A short statement on this would help clarify the role and complementarity of SAR within the dataset.

   Good point. We've added the following sentence the start of section **3.1.2**

   The ITS_LIVE project also includes velocity products derived from Synthetic Aperture Radar (SAR) imagery. SAR imagery has qualities that are valuable for imaging of polar glaciers and ice sheets as retrievals are not obscured by cloud or limited by solar illumination. These capabilities are highly complementary to optical retrievals.

1. Section 3.1.2 (Sentinel-1 processing) is relatively brief compared to the optical processing discussion. For example, while the use of a 21×21 Wallis operator is noted, there may be additional reasons beyond local variability in radar backscatter caused by topography. Is this choice optimized for a particular spatial resolution (e.g., 120 m in this case) or signal characteristic? Additionally, clarifying the resolution differences between the input datasets (optical vs. SAR) and how they are reconciled would improve reader understanding.

   We've added a sentence that points the reader to our previous publication on the SAR processing: "See Lei et al. (2022) for a more detailed description of the Sentinel-1processing.". We had not made it clear that a more detailed description of the processing has already been published. We've also added a new table (Table 1) that lists the characteristics of the source imagery so that it is now easier for the reader to identify differences in input imagery. As for the high-pass filter (21×21 Wallis operator)… unlike optical, SAR derived velocities are relatively insensitive to choice of high-pass filter owing to the speckled nature of SAR imagery. The only reason for its application is to protect against possible biases introduced by gradients in brightness due to topography.

1. The term "SLC" appears in two different contexts: "Scan Line Corrector failure (SLC-off)" on line 176 and "SLC (Level 1.1)" on line 203. Since the latter often

refers to Single Look Complex data in SAR terminology, this could confuse readers. Please consider clarifying the intended meaning in each case.

Good point. This isn't ideal. We've removed the acronym "SLC-off" as it was defined but not used again. We've also added a "Single Look Complex" definition for SLC when it is first used in the description of the radar processing.

1. Please ensure that all abbreviations are defined on first use, including: MEaSUREs, NISAR, AWS SNS, AWS SQS, and USGS STAC. While many readers may recognize them, others may not. For example, the NISAR acronym is explained in line 425, but it is first mentioned in line 106 — consider moving the definition earlier.

It looks like we were missing a lot of acronym definitions... we've reviewed the paper in full and made sure that the first occurrence of an acronym is proceeded by its definition.

---

## Author Response (AR3)

Dear Wesley,

Thank you for volunteering your time to be the editor of our paper and thank you for your suggestions. We have adopted all but your last suggestion to modify Figure 4 as the figure is generated from our online JavaScript web application [https://its-live.jpl.nasa.gov/app/index.html] and can only be exported as a PNG. Detailed responses below.

With much gratitude,

Alex Gardner and co-authors

*At line 46/47: Could you please also add GVT to this list? See reference here: https://tc.copernicus.org/articles/15/2115/2021/*

*Good suggestion, now included*.

Table 1: Could you add NISAR to this table? At the moment, this is anticipatory and this can be noted in the table, but as the paper is cited in the future, it would be good to have the expected NISAR datasets used in Its-Live be listed I think.

*Good suggestion, now included. Fingers crossed NISAR launches next month.*

Figure 4: Could you please make the fonts a bit larger on this figure, for the published version? Please also be aware of this recent publication: https://journalhosting.ucalgary.ca/index.php/arctic/article/view/80162/57726

*As explained above, editing the font size in this figure is problematic as the figure is produced from our JavaScrip web application. It's also kind of nice to have the figure exactly match what the reader would get if they clicked on the URL that is included in the figure caption [https://its-live.jpl.nasa.gov/app/index.html?z=11&lat=60.0151&lon=-139.4886&lat=60.0378&lon=-139.4213&lat=60.0548&lon=-139.3423&int=1&int=100&x=2017-11-10&x=2024-12-12&y=-31&y=5115]. It seems that the suggested paper is inaccessible. Maybe the ucalgary server is down. I was able to see the title of the paper and that it relates to Hubbard Glacier seasonal velocities. Given that this is just*

*an example plot for data exploration, I think that it's probably OK if we don't include citations to other works here.*